# A Systematic Review of the Effect of *Centella asiatica* on Wound Healing

**DOI:** 10.3390/ijerph19063266

**Published:** 2022-03-10

**Authors:** Elena Arribas-López, Nazanin Zand, Omorogieva Ojo, Martin John Snowden, Tony Kochhar

**Affiliations:** 1School of Science, Medway Campus, University of Greenwich, Central Ave, Gillingham, Chatham Maritime, Kent ME4 4TB, UK; e.arribaslopez@greenwich.ac.uk (E.A.-L.); n.zandfard@greenwich.ac.uk (N.Z.); m.j.snowden@greenwich.ac.uk (M.J.S.); 2School of Health Sciences, Avery Hill Campus, University of Greenwich, Avery Hill Road, London SE9 2UG, UK; 3HCA London Bridge Hospital, Tooley Street, London SE1 2PR, UK; tonykochhar@gmail.com

**Keywords:** *Centella asiatica*, burn, cytokine, collagen, contraction, wound granulation, re-epithelialization, wound healing

## Abstract

Background: Under metabolic stress conditions, there is a higher demand for nutrients which needs to be met. This is to reduce the risk of delay in wound healing which could lead to chronic wound. Aim: This is a systematic review of the effect of *Centella asiatica* on wound healing. *C. asiatica* is a traditional medicinal plant used due to its antimicrobial, antioxidant, anti-inflammatory, neuroprotective, and wound healing properties. Methods: PRISMA (Preferred Reporting Items for Systematic Reviews and Meta-Analyses) guidelines were followed for the systematic review and four electronic databases were used. Results: Four clinical trials met the inclusion criteria. The following distinct areas were identified under *C. asiatica*: wound contraction and granulation; healing/bleeding time and re-epithelialization; VAS (visual analogue scale) scores; skin erythema and wound appearance. Conclusions: *C. asiatica* might enhance wound healing resulting from improved angiogenesis. This might occur due to its stimulating effect on collagen I, Fibroblast Growth Factor (FGF) and Vascular Endothelial Growth Factor (VEGF) production. Besides, *C. asiatica* has shown an anti-inflammatory effect observed by the reduction in Interleukin-1β (IL-1β), Interleukin-6 (IL-6) and Tumour Necrosis Factor α (TNFα), prostaglandin E2 (PGE2), cyclooxygenase-2 (COX-2), and lipoxygenase (LOX) activity. Delivery systems such as nanoencapsulation could be used to increase *C. asiatica* bioavailability. Nevertheless, more studies are needed in order to perform a meta-analysis and ascertain the effects of *C. asiatica* on wound healing and its different parameters.

## 1. Introduction

A wound is known as the physical break of functional tissues [1,2,3,4]. Healing, on the other hand, begins right after an injury [3,5,6,7,8,9] and consists of four sequential stages. These phases are overlapped and can persevere for years [5,10,11,12,13] (Figure 1). The healing process is not linear, and it can move backwards and forwards through the stages depending on extrinsic and intrinsic factors, including cytokines and growth factors, among others. As a consequence, various interventions including nutritional interventions [14] have been developed with the purpose of promoting the wound healing process. Furthermore, there has been growing interest amongst researchers in the use of natural products to stimulate wound healing, for instance, *Centella asiatica* extracts.

### 1.1. Centella asiatica

*Centella asiatica*, also known as Gotu Kola, Bua-bok, Tiger grass, or Indian Pennywort [15,16], is an herbaceous perennial plant member of the *Apiaceae* family, also known as *Umbelliferae*. It has an important traditional value, particularly in South East Asia, due to its nutritional and therapeutical properties. It has been recommended for the treatment of a variety of skin conditions such as lupus, eczema, psoriasis, leprosy [17], and varicose ulcers [18]. Indeed, *C. asiatica* is widely used owing to its antimicrobial, antioxidant, anti-inflammatory, neuroprotective, and wound healing properties [19,20]. *C. asiatica* extracts have been shown to positively influence wound healing by improving collagen synthesis [18,21] and microcirculatory function [22,23].

*C. asiatica* extract has also been traditionally used for the management of keloids. Keloids are caused by the higher proliferation and deposition of collagen, also known as fibroproliferative lesions. This occurs due to abnormal healing and causes collagen proliferation beyond the wound margins [24]. Transforming growth factor- β1 (TGF-β1) plays a role in this pathology, as it induces collagen formation, while plasminogen activator inhibitor 1 (PAI-1) prevents the action of plasminogen activators. These are responsible for the dissolution of clots, which seal blood vessels, protecting them and preventing blood loss. Therefore, the inhibition of both TGF-β1 and PAI-1 by *C. asiatica*, mainly by one of its constituents, asiatic acid, makes it a promising compound for the successful management of keloids.

The main triterpenes found in *C. asiatica*, also known as centelloids, are asiatic acid (AA), asiaticoside (AS), madecassoside (MS) or brahminoside, and madecassic acid (MA) or brahmic acid [25,26,27]. The amount of these bioactive compounds in the plant depends on the tissue and ecotype used for their extraction, whether alcoholic or aqueous [28]. Both the oral and topical treatment of the alcohol extract of *C. asiatica* have been shown to stimulate synthesis, maturation, and the crosslinking of collagen in mice [29]. This enhanced healing was also observed in diabetic guinea pigs after the application of a cream containing 0.4% of asiaticoside in punch wounds. This occurred due to the higher stimulation of collagen and hydroxyproline synthesis in the wounds.

The enhancement of wound healing, particularly angiogenesis [30,31] might be due to the stimulation of collagen I, as well as the increased production of Fibroblast Growth Factor (FGF) and Vascular Endothelial Growth Factor (VEGF). This occurs via the activation of the TβR2 kinase-independent pathway [32,33]. In early angiogenesis, FGF promotes the proliferation of endothelial cells. Likewise, VEGF contributes to the formation of new capillaries through the regulation of cell proliferation, differentiation, and migration [34]. Moreover, VEGF stimulates vasodilatation and the formation of the extracellular matrix [35].

*C. asiatica* has shown an anti-inflammatory effect observed by the reduction in Interleukin-1β (IL-1β), Interleukin-6 (IL-6) and Tumour Necrosis Factor α (TNFα) [33,36,37], as well as prostaglandin E2 (PGE2) [36,38], and cyclooxygenase-2 (COX-2) [39]. In addition, *C. asiatica* has been shown to reduce inflammation by inhibiting lipoxygenase activity and lessening proteinase activity, thus inhibiting protein denaturation [40]. This is important since the suppression of protein denaturation can improve rheumatoid arthritis [41].

One of the current challenges with *C. asiatica* is the low bioavailability of its active compounds (mainly AS, AA, MS and MA). As a consequence, several delivery systems are being studied in order to increase their bioavailability. Some of these delivery systems include the use of nanoparticles [42], nanofibers [43], hydrocolloids or hydrogels [44], and nanoencapsulation [45,46,47,48]. Different studies have shown that *C. asiatica* properties could be enhanced using nanoencapsulation. When using nano lipid carriers (NLC), it has been shown that AS penetrated skin layers when applied topically [47]. Moreover, its biological activities could be increased by up to 50–60% owing to an enhanced bioavailability [45,48].

### 1.2. Why It Is Important to Perform This Review

There have been several trials studying the effect of *C. asiatica* on Chronic Venous Insufficiency, and a review addressing it can be found [49]. There have also been a number of articles regarding the effect of *C. asiatica* on wound healing, both in vitro [50,51] and in vivo [33,52,53,54,55]. However, there is currently a lack of clinical trials regarding its effect on wound healing, or any of the parameters affecting wound healing on patients. Furthermore, it would appear that no systematic review or quantitative synthesis have been conducted in this area of research and practice; these are needed to provide an evidence base for researchers and practitioners.

Aim:

The aim of this systematic review is to evaluate the effectiveness of *C. asiatica* on the promotion of wound healing in humans.

## 2. Methods

This systematic review was performed according to the Preferred Reporting Items for Systematic Review and Meta-Analysis (PRISMA) statement [56]. Furthermore, the review follows the Population, Intervention, Comparison, and Outcome (PICO) characterization.

### 2.1. Search Strategy

The following databases were searched for relevant papers: Pubmed, Science Direct, Medline, grey literature research with Google Scholar.

Based on the search strategy, the following keywords were used: *Centella asiatica*, Gotu Kola, Bua-bok, healing, wound, skin, cytokines, interleukin, inflammation. Words were combined using Boolean operators (OR/AND) (Table 1). References from pertinent articles were also examined for additional studies. Searches were conducted and data from the selected articles were extracted by one researcher (E.A.L.) and cross-checked by another researcher (N.Z.).

### 2.2. Study Selection

Inclusion criteria: The studies included in the review were randomised controlled trials (RCTs) conducted on adults treated with *C. asiatica*. These patients were healthy or unhealthy and suffered from either chronic or acute wounds (Figure 2).

Exclusion criteria: The studies excluded from the review were: those not conducted on humans; those not including supplementation or topical treatment with *C. asiatica* or any of its extracts; and those involving participants under 14 years of age. The reason for excluding studies with participants younger than 14 years is due to the metabolic stress that is present during growth. Studies in another language other than English, French, or Spanish, or with non-original or a lack of data, were also excluded from the review. No publication date restrictions were applied.

#### 2.2.1. Population

Patients were healthy or unhealthy adults who were over 14 years old, suffering from either chronic or acute wounds.

#### 2.2.2. Intervention

Oral or topical treatment with *C. asiatica* for at least 3 weeks.

#### 2.2.3. Comparator

A control group, which was either placebo-treated or untreated.

#### 2.2.4. Outcomes

The outcomes included in the review were: wound contraction and granulation (%); healing/bleeding time and re-epithelialization (days); VAS (visual analogue scale) scores; skin erythema and wound appearance.

#### 2.2.5. Data Extraction and Management

Data were extracted from figures using WebPlotDigitizer (Rohatgi, A., Pacifica, CA, USA) [57], tables, and the test from the articles.

#### 2.2.6. Quality Assessment

The risk of bias assessment was assessed by the Cochrane risk of bias tool (The Cochrane Collaboration, Copenhagen, Denmark) [58]. The domains evaluated included the random sequence generation (selection bias), allocation concealment (selection bias), blinding of participants and personnel (performance bias), blinding of outcome assessment (detection bias), incomplete outcome data (attrition bias), selective reporting (reporting bias), and other bias. Low risk of bias is indicated by a plus (+), unclear risk of bias by a question mark (?), and high risk of bias by a minus (−).

## 3. Results

Four studies [59,60,61,62] on *C. asiatica* were included in the systematic review (Table 2).

### 3.1. Assessment of Risk of Bias of Included Studies

The risks of bias summary in the included studies on *Centella asiatica* are shown in Figure 3. Notably, 100% of the studies showed a low risk of bias concerning the random sequence generation, allocation concealment, and incomplete outcome data. On the other hand, 50% of the studies demonstrated a low risk of bias with respect to the blinding of participants and personnel. The exception for this bias comes from the high bias in the case of Chiaretti et al. [61] and the unclear risk of bias in the case of Paocharoen [59]. In terms of blinding of outcome assessment, all the studies showed a low risk of bias except for Paocharoen’s study [59], which showed an unclear risk of bias in this parameter. Concerning selective reporting, all the studies showed a low risk of bias except for Saeidinia et al. [60], which presented an unclear risk of bias. Regarding other risks of bias, 50% of the studies showed an unclear risk of bias [61,62], whereas the other half showed a low risk of bias [59,60].

### 3.2. Effects of Interventions

Based on the systematic review, various distinct areas were identified under *C. asiatica* treatment: wound contraction and granulation; healing/bleeding time and re-epithelialization; VAS (visual analogue scale) scores; skin erythema and wound appearance.

### 3.3. Centella asiatica Extracts

*C. asiatica* has been shown to positively influence several factors involved in wound healing both in in vitro and in vivo [67] studies. Its beneficial effect on wound healing has also been confirmed in some clinical trials, which are reported below. In these trials, patients suffering from either an acute or a chronic wound were treated either orally or topically with different forms of *C. asiatica*.

#### 3.3.1. Wound Contraction and Granulation

In a clinical trial carried out by Paocharoen [59], 170 patients suffering from diabetic wounds were treated with two oral capsules containing either 50 mg of asiaticoside or a placebo, three times a day. It should be noted that the authors did not mention what they used as a placebo. As a consequence, it cannot be guaranteed that the placebo used did not have an effect on wound healing. The effect of supplementation on wound contraction and granulation were recorded on days 7, 14 and 21 after wounding. Wound contraction was assessed as the decrease in the volume of the wound, whereas wound granulation was evaluated as the decrease in the depth of the wound. Compared to the control group, the study group presented a significant increase in wound contraction in all the time points studied (*p* ≤ 0.001). Regarding wound contraction, both the *C. asiatica* (CA) group and the control group showed good wound contraction on days 7 (28.57 vs. 12.79%; *p* = 0.001), 14 (38.10 vs. 18.60%; *p* < 0.001) and 21 (57.14 vs. 44.19%; *p* < 0.001), although the contraction was more pronounced in the study group compared to controls. Due to the initial differences in wound depth and size between the groups, this trial studied the change in volume and area of the wounds in order to be able to objectively compare the changes in the wounds.

With respect to wound granulation, 42.86% of patients in the CA group vs. 16.28% in the control group presented no granulation tissue forming on day 7 (*p* < 0.001); 14.29 vs. 3.49% on day 14 (*p* < 0.001); and 9.52 vs. 3.49% on day 21 (*p* < 0.001) for CA and the control groups, respectively. These results suggest that *C. asiatica* might inhibit tissue overgrowth. A reason for this could be the anti-inflammatory effect and the regulation of collagen synthesis.

#### 3.3.2. Healing Time and Re-Epithelialization

Two trials [60,61] measured the effect of *C. asiatica* treatment on healing time.

Chiaretti et al. [61] supplemented 98 patients suffering from chronic anal fissure with 60 mg tablets of CA twice a day for 15 days, and compared the results to a control group who only received the traditional treatment. The group supplemented with CA showed a shorter healing time (MD: 3 weeks, 95% CI 2–3 weeks) compared to the control group, who needed an average of 4 weeks to heal (95% CI 4–5 weeks), although this difference did not reach significance (*p* = 0.07). In this study, the healing time was determined by the time needed for the wound to stop bleeding. This method of measuring healing time might differ from other studies, since it can also be measured as the time that the wound needs to be fully closed.

Saeidinia et al. [60] treated 75 patients suffering from second-degree burns. These patients were topically treated once a day, with either Centriderm (a topical ointment containing CA) or 1% silver sulfadiazine cream. The latter is normally used as a standard treatment; hence it was used as a control in this study. The study group healed quicker than the control group, needing an average of 14.67 ± 1.78 days, compared to the 21.53 ± 1.65 days required by the control group (*p* = 0.001). Besides, the meantime for re-epithelialization in the CA group was also lower, namely 13.7 ± 1.48 days compared to 20.67 ± 2.02 days in the control group (*p* < 0.0001). A reason for this outcome might be the effect of *C. asiatica* on the production of VEGF which, in turn, is stimulated by the production of monocyte chemoattractant protein-1 (MCP-1) by keratinocytes and IL-1β by macrophages, thus accelerating wound healing.

It should be noted that this is the only study where patients of an age lower than 18 years old were included in the study. The inclusion criteria of this study started at 14 years of age, which differs from the usual criteria used in adult clinical trials.

#### 3.3.3. VAS Scores

Chiaretti et al. [61] reported a reduction in VAS scores in people suffering from chronic anal fissure in the study group compared to the control group after one week of treatment (3.94 ± 1.62 vs. 5.37 ± 1.92 VAS scores). However, the difference between groups did not reach statistical significance (*p* = 0.07) until the second week of the study (3.64 ± 1.92 vs. 2.16 ± 1.03 VAS scores, *p* < 0.035).

On the contrary, a significantly favourable effect on VAS scores was observed from day 3 in another study conducted by Saeidinia et al. [60] on burn patients. The VAS scores in the CA group and in the control group were 1.70 ± 2.46 vs. 5.60 ± 1.63 on day 3; 0.17 ± 0.64 vs. 3.27 ± 1.70 on day 7; and 0.07 ± 0.35 vs. 1.17 ± 0.95 on day 14, (*p* = 0.001), respectively. Vancouver Scar Scale (VSS), including pliability, pigmentation, height and vascularity of the wound, was also studied in this trial. All the parameters were lower in the study group compared to the control group (*p* = 0.001). Nevertheless, pigmentation did not reach statistical significance until day 14 of the treatment (*p* = 0.001).

#### 3.3.4. Skin Erythema and Wound Appearance

In a recent trial carried out by Damkerngsuntorn et al. [62], 30 patients suffering from acne were treated on both sides of their faces with either a gel containing 0.05% of ECa 233, or the same gel without the active compound (ECa 233). ECa 233 is an extract of *C. asiatica* containing a standardised amount of triterpenes, namely 53.1% of madecosside and 32.2% of asiaticoside [68]. The treatment was carried out four times a day for 7 days. After the treatment, the skin treated with *C. asiatica* showed improved erythema (*p* = 0.009, 0.0061, 0.012) and wound appearance at days 2, 4, and 7. These differences reached a statistical significance of *p* = 0.008, 0.001, and 0.044, respectively.

## 4. Discussion

*C. asiatica* is commonly used in South East Asian culture for the treatment of lupus, leprosy, eczema, psoriasis [17], and varicose ulcers [18] due to its medicinal properties. There is a range of clinical trials studying the effect of *C. asiatica* on wound healing and the parameters influencing it as well as its mechanisms of action. Nevertheless, the findings of this review have revealed that *C. asiatica* might have a beneficial effect on wound healing. This favourable effect is observed with a quicker wound contraction, probably owing to the stimulation of fibronectin [69] and collagen I synthesis [21,31] and matrix remodelling [31]. These two are characteristic of the proliferative stage of the wound healing process [70,71]. Furthermore, *C. asiatica* has been found to be efficacious in the maintenance of connective tissue [72] and the strengthening of weakened veins [73]. As a consequence, its use might be beneficial for the treatment of venous insufficiency [63] as well as hypertensive microangiopathy [64]. Its oral supplementation has been shown to increase collagen synthesis and cellular proliferation [36], in addition to fibroblast division [52] after injury. It might also enhance the wound breach power in incision models (*p* < 0.001) as well as re-epithelialization [66] and wound contraction [23]. *C. asiatica* has also been shown to improve the tensile strength of the newly formed skin of the wound [29,74] in animal studies, which could lead to a decrease in the wound area and faster healing [30,75]. The promoting effect on collagen synthesis has already been suggested in a study where different cancer cell lines were treated with 50 mg/mL of CA extract [76]. In this study, a three-fold increment in collagen synthesis was observed. Similarly, the enhancement in collagen synthesis was, in a later study, performed on human dermal fibroblasts (HDFs). In this case, cells were treated with 30 μg/mL of asiaticoside, and the expression of collagen was improved after 2 h of treatment and for 48 h [77]. In another study performed on guinea pigs, both oral and topical treatment showed increased synthesis, maturation and the crosslinking of collagen. Indeed, following topical 0.2% asiaticoside application, increases of 56% and 57% were observed in hydroxyproline (*p* < 0.001) and tensile strength (*p* < 0.05), respectively, thus, resulting in enhanced re-epithelialisation and wound healing [29]. Similar results were observed when treating diabetic rats with 0.4% of topical asiaticoside.

The increase in collagen I synthesis in wound healing has been previously demonstrated by increased tensile strength in in vivo studies [74]. This increment might be due to the pro-angiogenic effect due to the increase in VEGF and FGF. Both growth factors play an important role in the wound healing process, mainly in the haemostasis and proliferation and repair stages, hence influencing wound healing [31]. Likewise, during angiogenesis, VEGF regulates cell proliferation, differentiation, and migration [35]. This promotes the formation of new capillaries [34], which allows an improved circulation to the wound site, hence providing essential nutrients and oxygenation [78,79,80]. The mechanism by which VEGF is stimulated results from an increased expression of some mediators, IL-1β and Monocyte Chemoattractant Protein-1 (MCP-1) [54]. Both IL-1β and MCP-1 also recruit macrophages, which help suppress inflammation and coordinate tissue repair [81]. On the other hand, FGF has also been proven to promote angiogenesis as well as stimulate the proliferation of endothelial cells in the wound site [82,83].

In addition, *C. asiatica* extracts might enhance wound healing time, re-epithelialization and wound appearance, assessed, in part, by wound pigmentation. These wound healing properties appear particularly effective when the extract contains greater amounts of AS [59], being one of its more active compounds. In addition, the enhanced proliferation of fibroblasts and the synthesis of the extracellular matrix has been reported after treatment with AS [84]. Asiaticoside has been shown to prevent the formation of keloids and hypertrophic scars by increasing the activity of immature collagen and myofibroblasts [31].

The healing process could be enhanced by treatment with *C. asiatica*, particularly in cases of persistent inflammation, as with the case of chronic wounds. This might result from the anti-inflammatory effect caused by *C. asiatica* due to the reduction in IL-1β, IL-6 and TNFα [33,36,37], as well as prostaglandin E2 (PGE2) [36,38] and cyclooxygenase-2 (COX-2) [39]. IL-1β, IL-6 and TNFα are pro-inflammatory cytokines that are secreted by inflammatory cells [85,86,87]. Likewise, inflammatory cells, including macrophages, can produce COX-2, a mediator of PGs. COX-2 is a catalyst in the conversion of arachidonic acid to PGE2 [88]. PGE2 plays a role in the regulation of the immune response and blood pressure [89]. As a consequence, during inflammation, PGE2 might lead to swelling, redness, and pain in the wound area. These symptoms occur due to increased blood flow to the inflamed area [90]. Consequently, a reduction in these mediators by *C. asiatica* might result in a lower inflammation in the affected area [91]. In addition, *C. asiatica* has been shown to reduce inflammation by inhibiting lipoxygenase (LOX) activity [40]. LOXs are key enzymes in the production of leukotrienes, which play a role in inflammatory diseases such as asthma, cancer, and arthritis [92].

Moreover, *C. asiatica* might be beneficial as a treatment of rheumatoid arthritis, as demonstrated in induced CIA mice, a condition that bears a resemblance to rheumatoid arthritis. In this case, the oral supplementation of MA resulted in the lessening of inflammatory cells in the joint areas rather than the inactivation of LPS-activated macrophages. This outcome suggests that its effect might be due to MA sapogenin or madecassic acid, and not MA itself, via cellular or humoral regulation [93]. This beneficial effect on rheumatoid arthritis has also been suggested due to its inhibitory effect on proteinase activity [41].

Delivery systems such as nanoencapsulation could be used to increase *C. asiatica*’s wound healing properties, extending the areas that can be treated due to its higher bioavailability [46] and penetration through the skin [54]. Furthermore, its biological activities can be enhanced by up to 50–60% compared to non-encapsulated *C. asiatica*. This intensified effect could be observed even when the nano-capsules contained a 30% lower amount of the compound than the control [52]. In accordance with the previous study, a 40% increased antibacterial activity of *C. asiatica* extract has been shown when using electrospun gelatin/silver nanoparticles [48].

The current data suggest that the constituents of *C. asiatica* might have a boosting effect on wound healing, not only when all the triterpenes are present, but also when used independently.

Also of interest are the oral amounts used in the previous studies, which were 300 mg of AS or 120 mg CA, although 60-120 is the usual dose used in actual herbal medicine [18,49]. The daily recommendation of *C. asiatica* by the World Health Organisation (WHO) ranges from 330 mg to 680 mg of CA extract, three times a day [94].

Regarding its topical application, the studies included in this review used a concentration ranging from 0.05% to 3 of CA.

## 5. Limitations of the Review

Only four studies were included in the systematic review, and this may limit the wider application of the findings of the review. Nevertheless, due to the lack of studies reporting data on the same parameters, it was not possible to conduct a meta-analysis to assess the effect of *C. asiatica* in the outcomes reviewed. Therefore, more studies are required in this area of research in order to effectively evaluate the effect of *C. asiatica* on wound healing.

## 6. Conclusions

Based on the findings of the systematic review, *C. asiatica* treatment might result in better wound healing due to greater angiogenesis and its anti-inflammatory effect. Moreover, this anti-inflammatory effect may result in reduced swelling, redness, and pain in the wound area due to the lessening of PGE2 and other inflammatory factors. This also suggests a promising effect in the treatment of rheumatoid arthritis. Notably, overall effect could be increased with the use of novel delivery systems.

## Figures and Tables

**Figure 1 ijerph-19-03266-f001:**
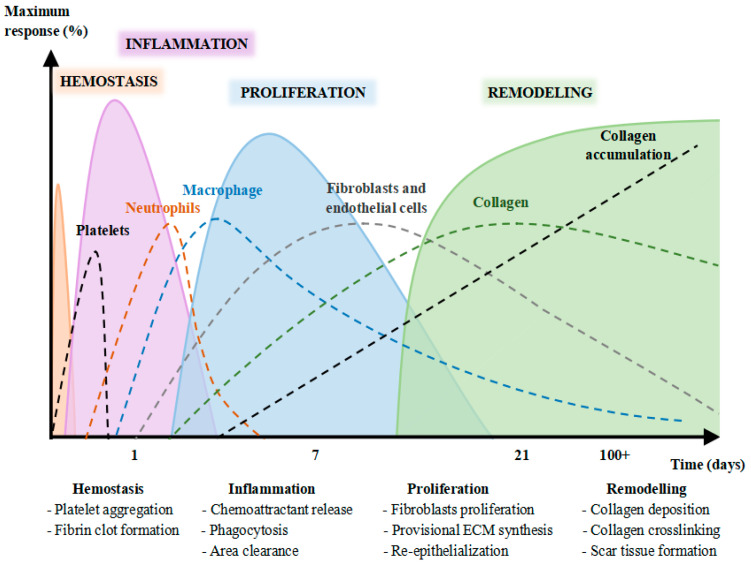
Stages of skin wound healing (haemostasis, inflammation, proliferation, and repair and remodelling) over time. Source: [14].

**Figure 2 ijerph-19-03266-f002:**
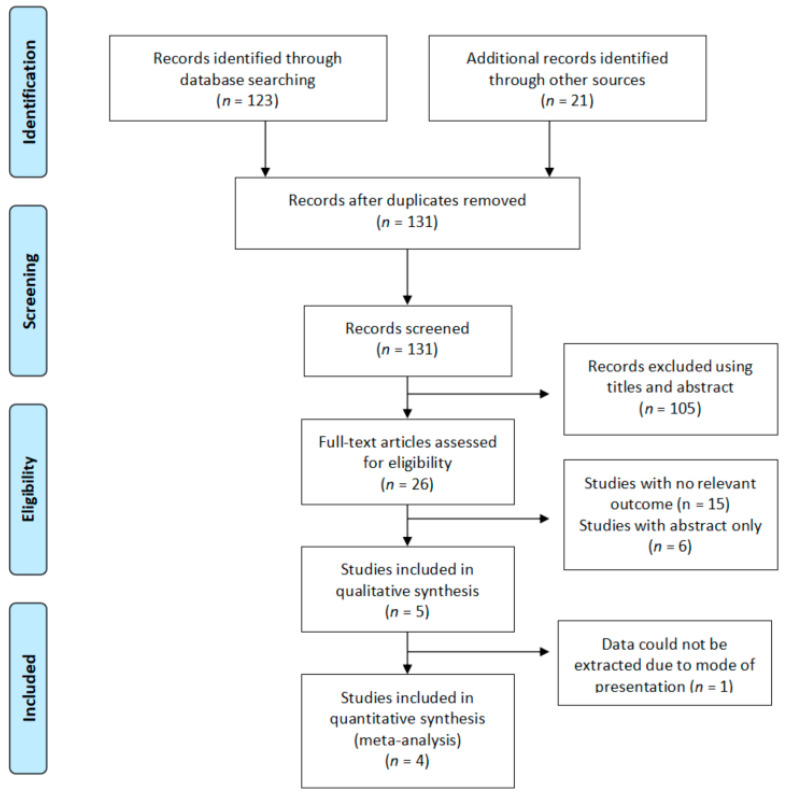
Flow diagram of the search strategy for *Centella asiatica*.

**Figure 3 ijerph-19-03266-f003:**
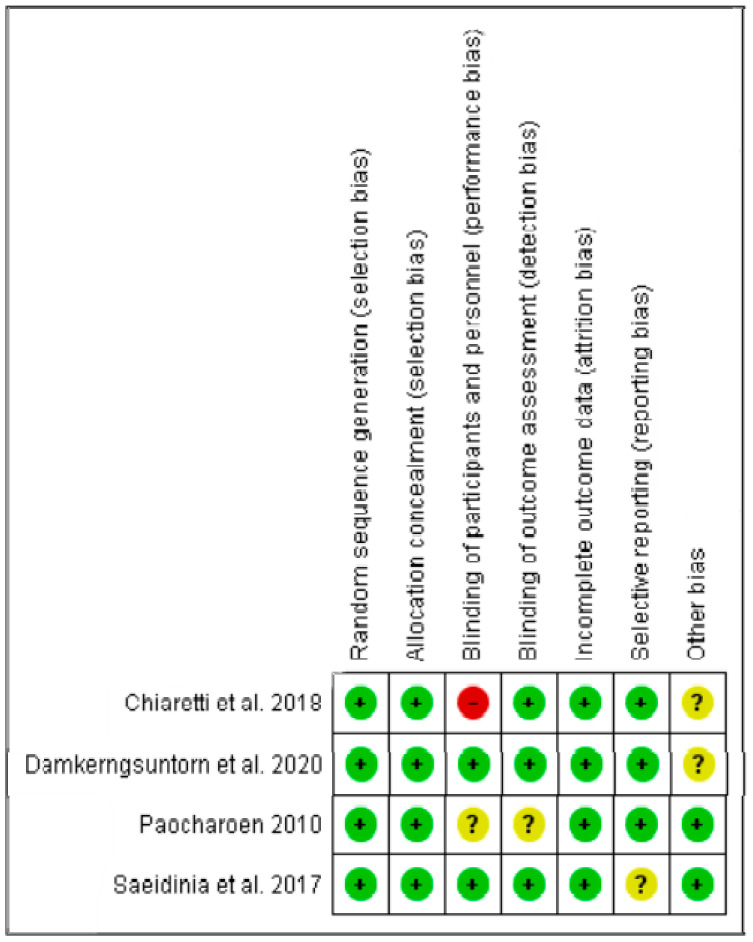
Risk of bias summary for the included studies on *Centella asiatica*. Low risk of bias (+), unclear risk of bias (?), and high risk of bias (−).

**Table 1 ijerph-19-03266-t001:** Search Terms and Search Strategy.

Patient/Population	Intervention	Outcome	Study Designs	Combining Search Terms
**Patients**			Randomised controlled trial	
**Diabetic wounds patients OR burn wounds patients OR acne treated patients OR chronic wounds patients**	*Centella asiatica* OR Gotu Kola OR Bua-bok	Inflammation OR Healing OR Wound OR Cytokines OR Interleukin OR Skin	Clinical trial OR Randomised controlled trial OR controlled clinical trial	Column 1 AND Column 2 AND Column 3 AND Column 4

**Table 2 ijerph-19-03266-t002:** Studies evaluating the effect of *Centella asiatica* treatment on wound healing and reported outcomes.

Study	Duration	Model	*n*	Compound	Control Group	Outcome
Paocharoen [63]	3 weeks	Diabetic wound patients	170	3 × 100 mg AS	Unspecified placebo	↑ Wound contraction, ↑ Wound granulation
Saeidinia et al. [64]	3.5 weeks	Burn wound patients	75	3% topical Centiderm	SSD	↓ VSS score, ↓ VAS score, ↑ Re-epithelialization, ↓ Healing time, Infection, ↓ Pigmentation
Chiaretti et al. [65]	8 weeks	Chronic anal fissure patients	98	2 × 60 mg oral + 3 g topical *C. asiatica*	Untreated	↓ Bleeding time, ↓ Pain (VAS scores)
Damkerngsuntorn et al. [66]	12 weeks	After laser treatment	30	Topical 0.05% ECa 233	Placebo	↓ Erythema, ↑ Wound appearance, ↑ Epithelialisation

AS: asiaticoside; ECa 223: 51% madecosside and 38% asiaticoside; SSD: Silver Sulfadiazine; TECA: Titrated Extract from *Centella asiatica*; TTFCA: total triterpenoid fraction of *C. asiatica*; VAS: visual acuity score; VSS: Vancouver Scar Scale; ↑: increases; ↓: decreases.

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
