# Peer review of "A Systematic Review of the Effect of Centella asiatica on Wound Healing"

_ijerph, 2022, doi:10.3390/ijerph19063266_

Round 1

Reviewer 1 Report

The manuscript investigated  “A systematic review of the effect on wound healing of Centella asiatica ” The findings in the paper are of significance in engineering and effect on wound healing. It is a meaningful work, but I think the manuscript still has alittle problems and needs to do more work to improve the quality of this manuscript to meet the standards for publication in IJERPH.  Under the present state, I therefore suggest a minor revision of the manuscript. I have added a detailed list of comments below: they should be taken into account by the authors when reworking the manuscript.

  1. The introduction are too long and concise.
  2. The following reference could support the discussion of this sentence.

Introduction of magnetic and supermagnetic nanoparticles in new approach of targeting drug delivery and cancer therapy application, ZM Avval, L Malekpour, F Raeisi, A Babapoor… - Drug metabolism reviews, 2020.

Bioactive agent-loaded electrospun nanofiber membranes for accelerating healing process: A review

SM Mousavi, ZM Nejad, SA Hashemi, M Salari… - Membranes, 2021

3- The English language must be carefully revised by a Native english Speaker.

4- References should be carefully rewritten.

5- The quality of Figure 2 is poor to be corrected.

6- Methodology. Examining the records you need to explain more.

Author Response

Thank you for your feedback. 

Please find attached a document with the changes done.

KInd regards

Reviewer 2 Report

The authors Elena Arribas-López et al. have reported the review article on the effect on wound healing of Centella Asiatica. The manuscript is written well and covered recent progress in the topic of interest. Henceforth, based on the merit of the manuscript, I would like to recommend the manuscript for publication. Though, before the publication, the following minor errors need to be improved.

  1. The term “Centella Asiatica” must be mentioned in italic throughout the manuscript.
  2. More clear figures must be included for better understanding
  3. The conclusion of the review must be written elaborately
  4. Very few references published in 2020 were cited. More recent references published in 2020-2022 should be cited and discussed.
  5. Some of the important references must be cited in a suitable place. Molecules 24 (7), 1437; Polymers 2020, 12(9), 2010; Prog Biomater. 2018 Mar; 7: 1–21; ChemMedChem 19 (5), 532 –544; Biomater. Sci., 2021, 9, 726-744..

Overall, the manuscript is sound worth publishing in the International Journal of Environmental Research and Public Health.

Author Response

(The authors gave the same response as above.)

Reviewer 3 Report

The authors demonstrated the application of Systematic Reviews and Meta-Analyses on wound healing activity of C. asiaticoside. This review is well-organized, readable, and potentially useful for the reader in the field and suitable for publish in International Journal of Environmental Research and Public Health. However, some modification should be added prior to publish.

  • Introduction part is well written.
  • In the methodology part, according to the review, there are only 4 papers found relevant to the search. Therefore, it might be useful to extend the systematic review on how we can improve wound healing properties of C. asiaticoside using other techniques, eg. delivery system or combined with other agents. In this case, authors could possibly collect more materials for the analysis and discuss the results.
  • Discussion and references, please explain and discuss more on the future direction of wound healing application of C. asiatica (eg. applying nanoencapsulation for enhancing it activity etc.) with more updated references.

Other minor points

  • The quality of the figure in this version was relatively low. The author should consider replacing them with high-resolution ones (Figure 1- Figure 3).
  • The abstract (background) should include some sentences on what CA is as well as its effects on wound healing.
  • There are a few errors on the link of reference, line 131, 169 etc.

Please check all abbreviations, they should be first mentioned with the full name, followed by (abbr.).

Author Response

Thank you for your feedback

Please find attached a ducoment with the changes done

Kind regards

Round 2

Reviewer 1 Report

 Accept in present form 

This manuscript is a resubmission of an earlier submission. The following is a list of the peer review reports and author responses from that submission.